# Physician-Led Thoracic Trauma Management in a Specialist Emergency Care Centre

**DOI:** 10.3390/jcm10245806

**Published:** 2021-12-11

**Authors:** Jonathan Bates-Powell, David Basterfield, Karl Jackson, Avinash Aujayeb

**Affiliations:** Respiratory Department, Northumbria Healthcare NHS Foundation Trust, Newcastle NE23 6NZ, UK; j.bates-powell@nhs.net (J.B.-P.); david.basterfield1@nhs.net (D.B.); karl.jackson@nhct.nhs.uk (K.J.)

**Keywords:** trauma, rib fractures, thoracic trauma

## Abstract

Introduction: Falls cause 75% of trauma in patients above 65 years of age, and thoracic trauma is the second commonest injury; rib fractures are the most common thoracic injury. These patients have up to 12% mortality, with 31% developing pneumonias. There is wide variation in care. Northumbria Healthcare has a team of respiratory consultants, physiotherapists, specialist nurses and anesthetists for thoracic-trauma management on a respiratory support unit. Methods: With Caldicott approval, basic demographics and clinical outcomes of patients admitted with thoracic trauma between 20 August–21 April were analyzed. A descriptive statistical methodology was applied. Results: A total of 119 patients were identified with a mean age of 71.1 years (range 23–97). Of the 119 patients, 53 were male, 66 females. The main mechanism of injury was falls from standing (65) and falls down stairs/bed or in the bath (18). Length of stay was 7.3 days (range 1–54). In total, 85 patients had more than one co-morbidity, 26 had a full trauma assessment and 75 had pan CTs. The mean number of rib fractures was 3.6 and 31 (26%) patients had a pneumothorax and/or haemothorax. A total of 18 chest drains were inserted (all small bore) and one needle aspiration was performed. No cardiothoracic input was required. Isolated chest trauma was present only in 45 patients. All patients had a pain team review, 22 erector spinae catheters were inserted with 2 paravertebral blocks. Overall, 82 patients did not require oxygen, 1 required CPAP and 1 HFNC. 7 needed intensive care transfer. Furthermore, 20 (17%) developed pneumonias and 16 (14%) deaths occurred within 30 days—all were in those with falls from standing. There was no correlation between number of fractured ribs, length of stay and mortality. Conclusions: High level care for thoracic trauma can be performed by a physician led team. Overall, 42% pneumothoraces/haemothoraces were observed. Further large scale randomised trials are warranted for definitive outcomes.

## 1. Introduction

There is an undeniable ageing of the population worldwide, and its effect is currently more marked in developed nations such as the United Kingdom (UK) [1]. In the UK, just under 250,000 hospital admissions, in patients 65 years old and older, were attributed to falls in 2018/2019 [2]. The North East of England, and more specifically the regions served by Northumbria Healthcare Foundation NHS Trust, has one of the highest rates [2].

The Northumbria Healthcare NHS foundation Trust is a large trust in the North East of England, and has a catchment area of approximately 600,000. It covers a large geographical area from Cumbria in the west towards the Scottish Borders. Care is organized across 4 main hospitals, and complimented with community hospitals. A specially commissioned acute care centre, the Northumbria Specialist Emergency Care Hospital, has just under 300 beds and was part of the National Health Service England’s New Models of Care Programme [3]. In this study, 24/7 specialist support cover was organized, and some aspects of patient care were re-structured. Major trauma is diverted to the regional trauma centre but a large proportion, approximately 95% (local data, unpublished) of trauma is seen locally. Historically, thoracic trauma was treated by the general surgical directorate, but care was shifted to the respiratory unit when the acute care centre opened. The unit provides ward-based high dependency care with 24/7 consultant cover, anaesthetics (incorporating the pain team) support and specialized physiotherapy. There is no local trauma centre, a regional trauma centre exists, and patients can be transferred there if required. As we are a district general hospital, in line with most of the United Kingdom, there is no on-site cardiothoracic service, but clear referral pathways exist to the regional centre. The service was ratified by the hospital’s department of medicine and surgery to cover medico-legal bases.

Studies from Global Burden of Diseases collaboratives point to injuries carrying significant mortality and morbidity all over the world, and road traffic accidents are the main contributor to injuries [4,5]. However, epidemiological data from the United Kingdom and the Trauma and Research Network (TARN), show that adults aged ≥65 years form the majority of patients presenting with major trauma. Falls from less than 2 metres (m) of height are the most common cause of injury, accounting for approximately 75% of those patients, and thoracic trauma accounts for approximately 50% of the injuries [6]. Overall mortality from trauma averages 10% but can be up to 50% in the elderly the most common reason being a pneumonia, in turn due to pain causing hypoventilation [6,7].

Management of thoracic trauma (which encompasses flail segments, rib fractures, haemothoraces, pneumothoraces and haemo-pneumothoraces) requires multi-disciplinary care. The core principles are appropriate triage, assessment, imaging, resuscitation, targeted individualized treatment of haemo-pneumothoraces and extra-thoracic injuries, careful monitoring, pain management and intensive physiotherapy [8]. There is wide variance in care [9]. National guidance is available [8]. Initial management decisions associated with thoracic trauma are made locally in the emergency department and then patients are triaged to the respiratory unit if appropriate or diverted to the regional trauma centre. National and international guidance suggests that all haemo-pneumothoraces should be drained [8,10,11] this is a controversial subject, and some evidence suggests that small (less than 2 cm as measured on computed tomograms) and/or asymptomatic haemo-pneumothoraces do not need to be drained and can be observed in a place of safety [12,13,14]. Drain size is left at the discretion of the treating physician, with the understanding that competency and confidence in the type of drain insertion (seldinger versus blunt dissection) is variable and dependent on resources [15].

Figure 1, presented below, shows the local decision pathway. (note COTE—care of the elderly, ED—emergency department, ESP—erector spinae catheter, PCA—patient controlled analgesia).

We reviewed our physician led high-level care and analyzed outcomes.

## 2. Methods

With local Caldicott approval, we collected data for a service evaluation on consecutive patients admitted with thoracic trauma to the respiratory unit between August 2020 and April 2021. An anonymized clinical record form was developed with Microsoft Excel 365. Continuous variables are presented as mean with ranges and categorical variables as percentages where appropriate. Simple descriptive statistics were performed. The size of all haemothoraces or pneumothoraces was assessed by a respiratory consultant.

## 3. Results

A total of 119 patients were identified. The average age was 71. 1 years (range 23–97). 53 (44.5%) were male and 66 (55.5%) female. Of these patients, 19 patients did not have any pre-existing medical conditions, 85 patients were multi-morbid, most commonly presenting with ischemic heart disease (present in 65 patients), hypertension (present in 59 patients) and chronic kidney disease (present in 42 patients).

The mechanisms of injury were falls from standing (6554.6%), falls down stairs/bed or in the bath (18), ladders (4), cycling (12), assault (3), road accidents (8) and 9 others (for example off horses). Figure 2 shows the percentage of the mechanisms of injury. The average length of stay was 7.3 days (range 1–54). And 85 patients had more than one co-morbidity.

Locally, a trauma assessment is defined as the correct use of a trauma document, an assessment incorporating all aspects of a trauma pathway (primary and secondary surveys), or the patient undergoing a trauma computed tomogram (CT), dependent on the clinical assessment. Injury severity scores are not calculated routinely.

The average number of rib fractures for any patient was 3.6 (range 1–10). All had a pain review team, 22 erector spinae catheters were inserted with 2 paravertebral blocks. Additionally, 82 patients did not require oxygen, 1 required continuous positive airways pressure (CPAP) and 1 high flow nasal cannula (HFNC).

Figure 3 shows the length of stay by number of rib fractures (X axis) and no correlation was observed (Spearman’s correlation coefficient 0.23, Z score 2.53, 95% CI 0.545, 0.4047, with similar results if the outliers are removed).

Isolated chest trauma was present only in 45 patients, 20 (17%) developed pneumonias and 16 (14%) deaths occurred within 30 days (1 heart failure and cancer progression, 2 COVID and 14 pneumonias), and all of these patients experienced falls from standing. The other injuries included sternal, scapular, pubic rami, vertebral, orbital, metatarsal, humerus and clavicular fractures as well as splenic, hepatic, adrenal and abdominal wall hematomas. Subdural hematomas were also present in three patients. All were managed on the respiratory unit, with specialist input from local orthopaedic, local surgical and regional neurosurgical teams. None of the patients required surgical intervention from general surgical and neurosurgical teams, and those (*n* = 4) who required orthopaedic surgery were managed pre-and post-operatively on the respiratory unit itself.

Of the patients, 31 (26%) had an isolated pneumothorax and/or haemothorax-20 pure pneumothoraces, 11 haemothoraces, and 6 (5%) had both. Of the 20 pure pneumothoraces, 17 were symptomatic from dyspnoea with large pneumothoraces, and the size of the pneumothoraces were large by established British Thoracic Guidance criteria [9]. A total of 18 chest drains were inserted (all small bore, defined as less than 20 French gauge). No cardiothoracic input was required as there were no prolonged leaks. Additionally, 1 haemothorax required a needle aspirate for symptom relief from dyspnoea. None of the other haemothoraces required intervention. No patients in the group whose pleural problems were simply observed required interventions down the line. The length of stay in the intervened group (those requiring pleural intervention) was 8.8 days (1–25 range) as compared to 6.75 days (2–12) in the observed (those not requiring pleural intervention) group; however, this did not reach statistical significance. {95% CI (−7.3629 to 3.2629), *p* = 1.0000}.

Patients over 65 years of age numbered 83, and 73 had more than one more comorbidity (3 with 1 comorbidity and 7 with none). Furthermore, 52 out of 83 had a full trauma assessment and 79 had falls from standing of less than 2 m in height. The average length of stay was 8.4 days, with a median of 6 (range 1–54, IQR 7), and incidence of pneumonia was 21 (25%). Appendix A compares those under 65 to the over 65. Values for length of stay, incidence of pneumonia and mortality did not reach statistical significance between the two groups.

Mortality was 19% in the over 65 s (overall 13%). The mean age of those who died was 84.9 years (range 75–97), and 47% (8) were male. The mechanism of injury was a fall from standing/less than 2 m in height in all. All had more than two co-morbidities (for example end-stage lung disease on long term oxygen in 3 patients, metastatic cancer in 4, heart failure in 12, dementia in 3 and various others). Nine had extra thoracic injuries (clavicle, pubic ramus, vertebral body, scapula, humerus fractures and a hepatic haematoma). Two had small isolated pneumothoraces, and one a small haemo-pneumothorax. None of the patients required pleural intervention. Three patients were positive for COVID-19 and died of respiratory failure, one developed a spontaneous intracranial hemorrhage on the ward, one developed progressive heart failure and kidney injury, one died of a rapidly progressive cancer, and one died due to significant frailty of old age. All the others developed bacterial pneumonias. Patient-centered ward-based ceilings of care were established, and palliative care provided when medical therapy failed.

## 4. Discussion

The above study shows that high-level care for thoracic trauma in an elderly co-morbid population can be performed by respiratory physicians and an anaesthetist-led acute pain team. Mortality is high in the over 65 s and this is heavily influenced by a number of unique patient variables such as the frailty of the population being admitted and mechanism of injury [7,16]. This is a unique set up in the United Kingdom, and has not been previously reported.

Thoracic trauma is anecdotally treated by surgical, anaesthetic or orthopaedic teams [8,9,10,11]. The importance of the physiotherapy and acute pain teams cannot be underestimated with regard to the success of this set up. In the absence of adequately powered randomized controlled trials on intervention, versus observation in traumatic haemo-pneumothoraces [17], good clinical acumen and patient centered decision making allows for the safe and effective management of this clinical problem.

The report above simply documents the outcomes of our service but shows that we deal with an increasingly elderly population that is multi-morbid, and our study confirms that falls from standing (less than 2 m in height) are associated with significant trauma. Our elderly patients are less likely to have a trauma assessment (the definition of which will vary in different hospitals) [7]. Our cohort is too small to infer any meaningful statistics.

However, the 19% mortality in the above 65 age group needs further explanation (overall mortality 13%). Previously described overall mortality averages 10–12% [7], and if we excluded the patients who died of COVID-19, a major cause of excess death, our mortality would be 10% overall, and 12% for the over 65 group. No patients below the age of 65 died, reflecting the fact that the above 65 group is frail and multi-morbid, often with irreversible, palliative pathology.

It has recently been proven in 585 patients (over the age of 65 years) with trauma from falls, that frailty, irrespective of its denominator and the age of the patient, directly correlates with survival at one year [16]. A frailty assessment could perhaps allow for a holistic approach and provide opportunities to improve outcomes. Our service does not have this, and would perhaps benefit from embedding frailty assessments: however, there is no data to suggest that embedding this would improve outcomes- further research is required with control groups.

Manual searches for patients with rib fractures and trauma were performed on the admission for the ward. This is one of our limitations, as well as the fact that we excluded patients managed on other wards. However, we still believe that the above-mentioned cohort is a representative sample. The main repository of data regarding trauma in the UK is the Trauma Audit and Research Network (TARN) [18]. TARN is the National Clinical Audit for traumatic injury and is the largest European Trauma Registry, holding data on >800,000 injured patients including >50,000 injured children. It is thus an extremely valuable source of information. However, TARN only accepts data from patients with traumatic injury with three or more overnight hospital stays or intensive care and/or high dependency care admission of any length or death due to injuries or if transfer to specialist major trauma centre is required. Unpublished data obtained for Northumbria Healthcare Foundation Trust spanning January to June 2021 reveals 347 patients, with an average length of stay of nine days, and average injury severity score (ISS) of 9. The vast majority of injuries (80%) were from falls less than 2 m in height. 107 out of 347 patients (31%) had a thoracic injury, and their length of stay was 7 days (same as above study), and the average ISS was nine. Again, falls from less than 2 m in height were the main causative factor. Overall, 88% of the thoracic injuries were isolated. Another limitation of our study is that we do not have a control group (for example those without thoracic injuries) and the TARN data, presented above, is the best we currently have. Further work is planned in this respect.

An important point to discuss is the conservative versus interventional management of traumatic haemo-pneumothoraces. Traumatic pneumothoraces occur in 20% of all trauma patients [10,12,13]. A systematic search of the available literature did not reveal any completed randomised controlled trials pertaining to intervention’s in chest trauma. There is one such study ongoing in Canada, which is a randomized controlled study comparing patients with traumatic haemothorax managed by intercostal drainage or expectant management [19]. A further trial is planned in the United Kingdom, Conservative Management in Traumatic Pneumothoraces in the Emergency Department (CoMiT-ED). A Randomised Controlled Trial, to determine if traumatic pneumothoraces can be managed with simple observation [20]. We have been invited to participate in this trial.

A significant educational drive is currently underway locally, regionally and nationally to increase awareness of the injury potential in the ‘elderly fallers’ [6]. This is a significant quality improvement project. Local radiological capacity will not sustain trauma-computed tomograms for all these patients, and the balance lies with good clinical acumen and a thorough assessment. Such a discussion is beyond the scope of this article, and we propose to submit a follow up to this article once we have implemented new systems and collected outcomes.

## 5. Conclusions

The physician-led management of thoracic trauma, complemented by specialized physiotherapy and anaesthetic-led pain team service, is feasible. Further service improvements are planned.

## Figures and Tables

**Figure 1 jcm-10-05806-f001:**
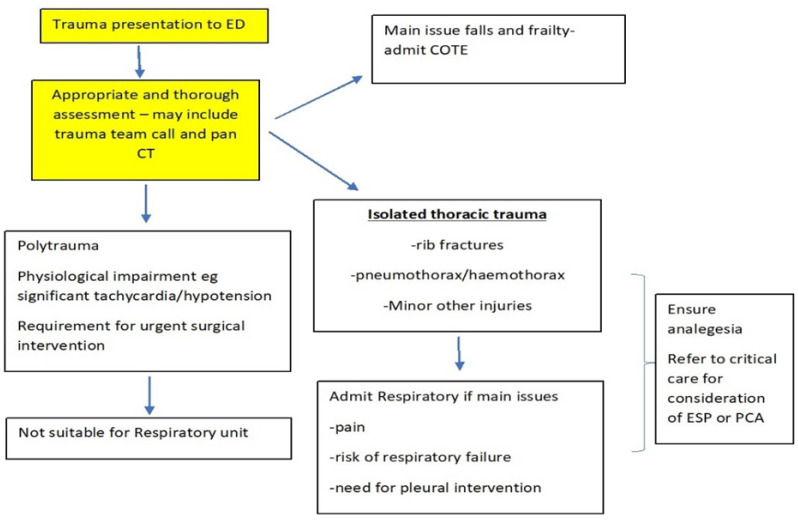
Local decision pathway for trauma presentation.

**Figure 2 jcm-10-05806-f002:**
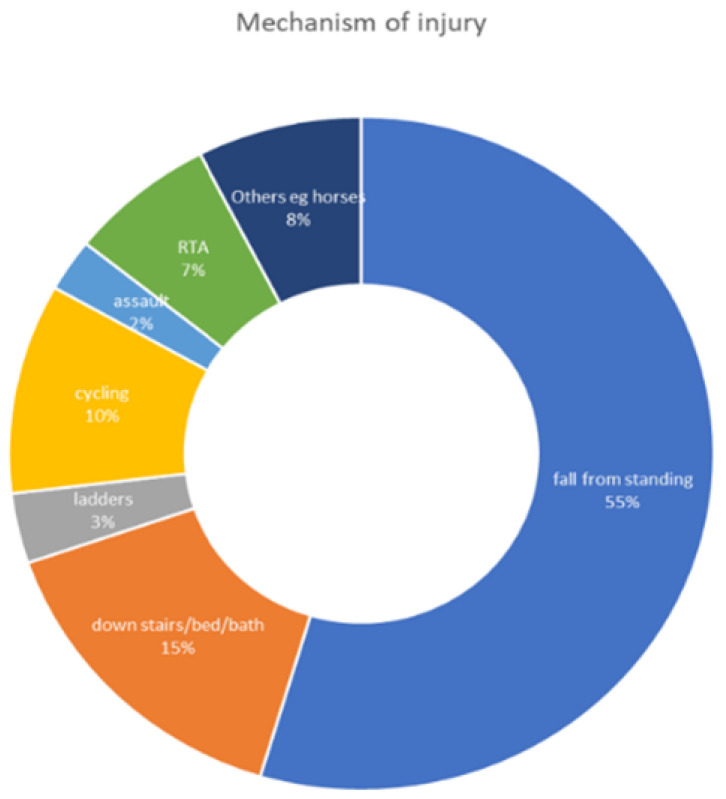
Percentage of the mechanisms of injury.

**Figure 3 jcm-10-05806-f003:**
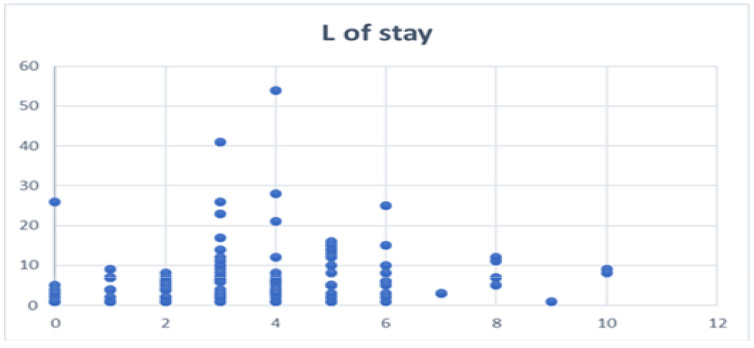
Length of stay by number of rib fractures.

## Data Availability

Data can be shared upon reasonable requests.

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
