# Peer review of "Physician-Led Thoracic Trauma Management in a Specialist Emergency Care Centre"

_jcm, 2021, doi:10.3390/jcm10245806_

Round 1

Reviewer 1 Report

I must say that as a thoracic surgeon this reviewer read with quite an interest the manuscript titled “Physician-led Thoracic Trauma Management in a Specialist Emergency Care Centre: Catering for the Older Faller” by Dr. Bates-Powell and colleagues from the Respiratory Department of the Northumbria Healthcare NHS Foundation Trust in Newcastle, United Kingdom.

I must admit that the notion of “physician lead trauma management” sounded to me as an oxymoron! However, I tried to approach this review, as we all must, unprejudiced and non-partisan!

I must say however, that reading the introduction of the manuscript and realizing the trauma management approach of the Northumbria Healthcare NHS foundation Trust left me astounded! Despite having left the U.K. for over 10 years now this reviewer is well aware of the NHS’s cost cutting driven initiatives which had started even at his time there, and which have continuously marginalized surgical specialties, have replaced surgeons with surgical assistants and nurse practitioners/prescribers and have reduced the surgical training positions nationwide.

However, to assign trauma management, and especially thoracic trauma to respiratory physicians is a stretch even to my imagination! In truth I am wondering how our chest physician colleagues have even accepted this themselfs, though the more and more I consider the drive within the respiratory medicine community to become more and more intervention oriented (i.e. performing pleuroscopy for pleural effusion or endobronchial valve insertions for LVRS ect, ect) it is I must admit plausible! In the end of course we will end up making respiratory medicine a surgical specialty.

The total lack of thoracic surgical input in the care pathway should be considered a malpractice and should be quite scary for our respiratory colleagues that they have been asked to take on a role which their specialty is not designed to cover; they have not been educated/taught about and have no experience on it!! Unless of course the U.K. courts have changed as well over the past 10 years.

Nevertheless, it is admirable that Dr. Bates-Powell and his colleagues have taken on this role and had the courage to present their experience and outcomes. For that I will commend them!

Therefore, in this retrospective, single center case series analysis spanning one year they present their management and outcomes of 119 patients with thoracic trauma.

The manuscript is well written, it is clear and concise and only requires minor spelling error corrections.

I do have a couple of issues which I would kindly like to be addressed:

The age range reported is from 23 to 97 years therefore the title is misrepresenting as care does not only focus on elderly patients which had falls. Presumably the younger population would have quite a different causative mechanism of trauma (possibly more intense and thus more debilitating), have a much better physiological profile (and thus better outcome) and therefore this introduces a significant range of bias in the analysis! Why did the authors not exclude these younger patients from their analysis?

Similarly the mechanism of injury, although mainly due to falls contains a significant proportion of other causes (RTA, cycling, assault, ect). These mechanisms can be more severe than a simple fall from standing therefore it again introduces another significant bias in the analysis.

In addition If isolated chest trauma occurred only in 45 patients does that not mean that there was also input from other surgical specialties (general surgery, orthopedics, neurosurgery, ect) and therefore the management of the majority of cases was not purely physician lead as hinted by the authors? This is actually admitted by the authors in line 115!

It is interesting that in 31 (26%) of patients which had a pneumothorax or/and a hemothorax “no cardiothoracic input was required”. Does that mean that no prolonged air leaks occurred or that the hemothoraxes were drained with a “less than 20 french” catheter?

The way the results are presented can be represented that only 2 mortalities occurred where as according to table 1, 16 deaths occurred in the group. That is almost 14 (13,4)% mortality which I am sorry to say it is not   “comparable to previously described evidence” and also should be the only factor the success of a service should be measured by!!! If a surgical service had a 14% mortality there would be an immediate investigation launched by the Healthcare Commission or the Healthcare Inspectorate!

I agree with the authors that all good trauma services should integrate good pain control/pain team input and physiotherapy in their armamentarium but surgeons have been doing this for decades and continue doing it with the advent of ERAS and other protocols! This is not something unique to this service and we don’t “need to emulate this set up”

In conclusion, this is a very controversial issue presented for publication. As mentioned the whole set up of the service is driven in this reviewer’s experience by cost cutting measures and in truth our physician colleagues willingly of forced are acting outside their mandate! Nevertheless, I would like to see this publication out following some revision because I believe it would stimulate discussion. Publication must however be accompanied by an editorial or a commentary by a thoracic trauma expert which I believe must raise the same issues I have.

I wish well to all.

Reviewer 2 Report

This study is not well designed and does not provide any results to conclude physician led high-level care would be associated with improved outcomes for patients only by descriptive statistical methodology. My comments are showed below.

  1. The study design is unclear. There is no comparison group in this study.

  1. The methods are too simple. How to select the study participants and how to collect the data.

  1. There are no any Inferential Statistics in this analysis and in methods of this manuscript.

  1. The format of citations in main text of this manuscript are errors, some are “upper index”, others are “[ ]”.

  1. Some points are typos

(1) “variables are presented as mean (±range)…” on line 82, the “±” should delete.

(2) The authors said “19 patients did not have any pre-existing medical conditions…” on line 87-88, however I found “Number with no comorbidities” were 17 patients (10 in “Age under 65 years” and 7 in “Age above 65 years of age” ) in Table 1; the same problem was found that “Number with more than 1 comorbidity” were 87 patients, but the authors said “85 patients were multi-morbid…” on line 88 and “85 patients had more than 1 co-morbidity.” on line 93-94.

6. “Falls from standing are associated with significant mortality and morbidity.” in abstract. It doesn’t make sense. I can’t believe that falls from standing are higher mortality than other mechanisms of injury. The authors should discuss in discussion.

Reviewer 3 Report

Authors have summarized that Physician-led management of thoracic trauma, complemented by specialized physiotherapy and anaesthetic-led pain team service is feasible, which will help physicians and clinicians to manage elder patients with thoracic trauma in the future. 

As a clinical report, though there is no control trial, authors have analysed and demonstrated the importance of physician-led management for trauma treatment in elderly patients, which will be helpful for other physicians. The manuscript is shown readable and well organized. Authors also analysed the causes of thoracic trauma, which may improve the healthcare. And they tried to make a decision pathway, which might be developed and updated by more physicians in the future. Thus, I support this manuscript.

Best, 

Reviewer 4 Report

Congratulation to the authors for this excellent paper

Physician-led Thoracic Trauma management in a specialist emergency care centre: catering for the older faller  

1) I consider the paper original or relevant in the field

2)add innovative aspect and perspectives to the subject area compared with other published

3)No suggestion for improvements should

- the conclusions are consistent with the evidence and arguments

4)the references are appropriate the tables and figures are really clear

Round 2

Reviewer 1 Report

Dear Editor and Dr. Bates-Powell and Colleagues,

I had the pleasure to re-review and re-consider your manuscript and I have read and taken into account your responses to my comments, which although they may sound harsh I do feel are fair and constructive.

I continue to remain skeptical about the whole concept of physician lead thoracic trauma. As I said in my previous report is sound like an oxymoron (like saying thoracic surgeon lead management of pneumonia!).

Nevertheless, I do give you the benefit of the doubt, i.e. that you were forced due to cost cutting measures/coerced by the trust/had to due to circumstances/regional setup ect (I hope you did not volunteer to take upon you this extra role outside your knowledge base and mandate) that you have tried to as our American colleagues so fondly say “tied to make lemonade out of lemons”. Therefore, as you have addressed the general manuscript issues I have raised and made appropriate corrections to your text I will support the publication of your work PROVIDED AN EDITORIAL – COMMENT BY A THORACIC TRAUMA SURGEON is published as well as I have previously mentioned. I presume as we all are open to discussion and feel comfortable with defending our position this would not be a problem (I note you are aggregable to this in your response comments).

I will therefore suggest it the editor and in truth I will lobby strongly for the publication of your work and for the commentary to stimulate discussion on the issue.

My kindest regards,

Dr. Emmanouil Kapetanakis

Author Response

Thank you. I did not mean to come across harsh, and if i did, I apologise. No specific comments are required for this round. Avinash

Reviewer 2 Report

  1. After reviewing this article catrfully once again. I think this article is a clinical report, it is not a research.
  2. The study design is still unclear. If this is an observational cohort study, the authors should define the exposure group and non-exposure group(the non-exposure group is also called comparison group). In addition, if this is an observational cohort study, why the authors said “The length of stay in the intervened group is 8.8 days (1-25 range) as compared to 6.75 days (2-12) in the observed group: this did not reach statistical significance” on line 133-135 in this revised manuscript. This would be an intervention study. So, what are the definitions for “the intervened group” and “the observed group”?
  3. The authors didn’t answer my question. My question was “There are no any Inferential Statistics in this analysis and in methods of this manuscript.” The authors said “We did not use that, but used simple descriptive methodology. This has been mentioned. We are simply mentioning the feasibility of our service. We have removed the part in the discussion where we have suggested that conclusions can be inferred.” However, the authors said “We hypothesized that such physician led high-level care would be associated with similar or improved outcomes for patients” on line 82-83 in this revised manuscript. In my opinion, if the authors want to test this hypothesis, they should use some inferential statistics, please consult the statistical experts. In addition, the authors said “Figure 3 shows the length of stay by number of rib fractures (X axis) and there was no correlation” on line 111-112 in this revised manuscript. Why was there no correlation? The authors should show the spearman’s correlation coefficient.
  4. I don’t know why the author want to present the Table 1. The author didn’t illustrate the meanings of Table 1 in Results. In addition, the data of “95% CI -7.3629 to 3.2629, Significance level P = 1.0000” were from “ Average number of rib fractures” or “Length of stay (days)”. Moreover, the authors said “The length of stay in the intervened group is 8.8 days (1-25 range) as compared to 6.75 days (2-12) in the observed group: this did not reach statistical significance. {95% CI (-7.3629 to 3.2629), P = 1.0000}” on line 133-135 in this revised manuscript. Why the data of “{95% CI (-7.3629 to 3.2629), P = 1.0000}”was the same with Table 1? Finally, I think the authors should add p value in each variable in Table 1, if the authors must present the Table 1.

Author Response

see notes
